# Total Flavonoids Extracts of *Apocynum* L. from the Ili River Valley Region at Different Harvesting Periods and Bioactivity Analysis

**DOI:** 10.3390/molecules27217343

**Published:** 2022-10-28

**Authors:** Deyi Shao, Gang Gao, Aminu Shehu Abubakar, Hanipa Hazaisi, Ping Chen, Jikang Chen, Kunmei Chen, Xiaofei Wang, Yue Wang, Yu Chen, Chunming Yu, Aiguo Zhu

**Affiliations:** 1Institute of Bast Fiber Crops, Chinese Academy of Agricultural Sciences, Changsha 410205, China; 2Department of Agronomy, Bayero University Kano, Kano P.M.B. 3011, Nigeria; 3Ili Agricultural Science Institute, Yining 835100, China; 4Key Laboratory of Biological and Processing for Bast Fiber Crops, Changsha 410221, China

**Keywords:** *Apocynum venetum*, *Apocynum hendersonii*, antioxidant, antimicrobial, flavonoids, metabolomics

## Abstract

In the current study, the total content from two *Apocynum* species leaves (*Apocynum venetum* and *Apocynum hendersonii*) collected from the Ili River Valley Region were extracted, and their bioactivities were investigated. The results showed a significant variation in the total flavonoid contents in the leaf samples collected at different periods (June, July, August, and September), with the highest content in August (60.11 ± 0.38 mg RE/g DW for *A. venetum* and 56.56 ± 0.24 mg RE/g DW for *A. hendersonii*), and the lowest in June (22.36 ± 0.05 mg RE/g DW for *A. venetum* and 20.79 ± 0.02 mg RE/g DW for *A. hendersonii*). The total flavonoid content was comparably higher in *A. venetum* than in *A. hendersonii*. Leaves extracts from the two species demonstrated strong bioactivity, which positively correlated with the total flavonoid contents. The anti-oxidative activity of *A. venetum* was higher than that of *A. hendersonii* in tandem with its higher flavonoid contents; the antibacterial activity, however, was conversely opposite. Furthermore, a total of 83 flavonoid metabolites were identified in the two species based on UPLC-ESI-MS/MS, out of which 24 metabolites were differentially accumulated. The variability in these metabolites might be the reason for the different bioactivities displayed by the two species. The present study provides insight into the optimal harvest time for *Apocynum* species planted in the major distribution area of the Ili River Valley and the specific utilization of *A. venetum* and *A. hendersonii*.

## 1. Introduction

Plant secondary metabolites that comprise polyphenols, alkaloids, and terpenes have been the source of pharmacology and therapeutics for decades [1]. Flavonoids are the most active secondary metabolites in controlling metabolic syndrome in vivo and in vitro [2]. They also exhibit a potential function in preventing chronic noninfectious diseases such as diabetes, cardiovascular disease and cancer [3]. Due to the health risks associated with consuming synthetic antioxidants, natural flavonoids extracted from plants are recommended, for they are relatively safe [4]. Flavonoids are known to alleviate reactive oxygen species and boost the body’s immune system, enabling the plant to have the most value in industrial use [5,6]. However, significant variations in total content, component, and bioactivity of natural flavonoids exist among plant species, varieties, and their geographical distribution [7]. Therefore, understanding the variations in natural flavonoids will bring a clearer guide for the precise utilization of plants. 

*Apocynum* species are perennial herbs belonging to the family Apocynaceae and were reported to be a unique source of polyphenols with medicinal significance [8]. Previous studies have detected a considerable number of flavonoid constituents in *Apocynum*, which were proven to impact stress resistance, antioxidant, antibacterial, anti-inflammatory, and sedative effects [9,10,11,12]. Extracts from the species alleviated doxorubicin-induced cardiotoxicity via the AKT/Bcl-2 signaling pathway and showed significant hepatoprotective effects against carbon tetrachloride-induced hepatotoxicity [13,14]. The phytochemicals characteristic made the plant adapt to drought and saline lands and a suitable alternative to synthetic human drugs for treating heart disease, hepatitis, and hypertension [15,16]. However, the total flavonoid content varies in different species and regions depending on their growing conditions and physiological distribution [7], which causes poor consistency of the natural extracts and derivative products. 

In China, *Apocynum* plants are mainly grown in Xinjiang Uygur Autonomous Region, especially in the area where agriculture is severely limited by soil salinity, alkalinity, and drought, owing to the plant’s excellent tolerance to stresses [17]. The Ili River Valley Region in the northwestern part of Xinjiang is the wettest [18]. Due to its unique topography, landform, and mountainous terrain, it has developed a humid temperate climate with abundant light, fertile soil, abundant water resources, and excellent natural conditions, which makes it suitable for the growth and distribution of *Apocynum* species [19]. Therefore, there is great significance in the comprehensive development and utilization of *Apocynum* genus plants in the Ili River Valley Region. The major cultivated species of *Apocynum* are divided into *Apocynum venetum* and *Apocynum hendersonii* (Figure 1A), which are excellent raw materials for both textiles and medicine, as well as pioneer crops for ecological restoration of saline soil and controlling desertification [8,20]. The biological activity of *Apocynum L.* has been associated with its rich flavonoids [21]. However, there are marked differences in total flavonoids and bioactivity between *A. venetum* and *A. hendersonii*, which require in-depth study to unravel the causes or stages.

In this study, the total flavonoid contents of the leaf extracts of *A. venetum* and *A. hendersonii* grown in the Ili River Valley Region at different harvesting stages were determined by the aluminum chloride colorimetric method. In addition, the antioxidant activities of the extracts at different harvesting stages were analyzed by DPPH, ABTS, and FRAP methods, and their antibacterial activity against *Escherichia coli*, *Staphylococcus aureus*, and *Aspergillus niger* was investigated. The composition and relative content differences of the metabolites of *A. venetum* and *A. hendersonii* were analyzed using a targeted metabolomics approach. This study provides new insights into the differences in total flavonoid and bioactivity between *A. venetum* and *A. hendersonii*.

## 2. Results and Discussion

### 2.1. Total Flavonoid Content Determination

The total flavonoid content of *A. venetum* and *A. hendersonii* leaves was determined at the four different harvesting stages. In previous studies, *A. venetum and A. hendersonii* contained significant amounts of flavonoids, and methanol was shown to be more effective in recovering flavonoids from the *Apocynum* [22]. Therefore, methanol was used as the extraction solvent, and the result of the total flavonoid content in the two species is shown in Figure 1B. There was significant variation in the total flavonoid content of the species at the different harvesting stages, and it was in the ranges of 20.79 mg RE/g DW to 60.11 mg RE/g DW, with an increasing trend in total flavonoid content from June to August and subsequently decreasing in September in both. *A. venetum*, however, displayed the highest amount of total flavonoids at the different harvesting stages compared to *A. hendersonii*. The highest total flavonoid contents of *A. venetum* and *A. hendersonii* leaves were both in the August harvesting period, with 60.11 ± 0.38 mg RE/g DW and 56.56 ± 0.24 mg RE/g DW, respectively. The lowest total flavonoid content was recorded in the June harvest (*A. venetum*, 22.36 ± 0.05 mg RE/g DW and *A. hendersonii*, 20.79 ± 0.02 mg RE/g DW). A different study showed the total flavonoid contents of *A. venetum* collected from Shanxi Province, China, in July 2006 and that of *A. hendersonii* collected from Qinghai Province, China, in July 2007 to have had 31.09 mg RE/g DW and 18.16 mg RE/g DW, respectively [23]. Both were lower than what was obtained in this study in the August harvest for *A. venetum* (60.11 ± 0.38 mg RE/g DW) and *A. hendersonii* (56.56 ± 0.24 mg RE/g DW). These might be due to the differences in the harvesting stage, climate, and extraction procedures. Accordingly, it can be concluded that August is the best harvesting period for *A. venetum* and *A. hendersonii* growing in the Ili River Valley Region, as this period has the highest total flavonoid content, which is reflective of medicinal significance.

### 2.2. Antioxidant Activity Analysis of Apocynum L. at Different Harvesting Stages

Due to the antioxidants’ different principles of action, diversity of substrates, active substances involved, and the complexity of the oxidation process itself [24], we used a combination of 2,2-diphenyl-1-picrylhydrazyl (DPPH), 2,2-azinos-bis (3-ethylbenzo thiazoline-6-sulphonic acid) (ABTS), and ferric-reducing antioxidant power (FRAP) to evaluate the antioxidant capacity of the various extracts.

DPPH is a stable free radical widely used to screen for active antioxidants in natural treatments [25]. The ability of *A. venetum* and *A. hendersonii* to scavenge DPPH free radicals was determined, and the result is presented in Figure 1C. The radical scavenging activities of *A. venetum* and *A. hendersonii* leaf extracts against DPPH free radicals ranged from 48.54% to 72.99% and 46.09% to 65.17% at different harvesting stages, respectively. There was a significant difference (*p* < 0.05) between the harvesting periods. The result followed similar trends to that of the total flavonoids, showing a substantial increase from June to August, before declining in the September harvest in each species. Between the two species, however, *A. venetum* extracts of all the harvest stages had comparably higher radical scavenging activity than its harvest pair from *A. hendersonii* (*p* < 0.05).

ABTS free radical solution reacts with antioxidants; the solution color fades in reflection of the antioxidant ability to scavenge free radicals [26]. The ABTS assay result for *A. venetum* and *A. hendersonii* ranged from 73.62% to 89.44% and 69.69% to 85.77%, as depicted in Figure 1D. The highest and lowest were in August and June, respectively, in each species, consistent with the DPPH result. Similarly, *A. venetum* leaf extracts displayed higher ABTS radical scavenging activity than *A. hendersonii* (*p* < 0.05) as in DPPH.

The Fe^3+^-TPTZ is reduced by the antioxidant to a stable blue Fe^2+^-TPTZ, where the color change reflects the antioxidant capacity of the reactants [27]. The reducing power of *A. venetum* leaf extracts, according to the FRAP result (Figure 1E), was higher than that of *A. hendersonii* leaf extracts and followed a similar pattern with DPPH and ABTS with the highest in the August harvest and the lowest in the June harvest.

Based on the three different antioxidant activities, both *A. venetum* and *A. hendersonii* leaf extracts had potentially excellent antioxidant activity in tandem with the relevant literature [1], though *A. venetum* displayed better capacity. The best harvesting time of the *Apocynum* is August when the target is the flavonoid contents. A study on *Tephrosia purpurea* that assessed the seasonal variation of the polyphenols and their antioxidant capacity reported significant variation in the antioxidant capacity of the three harvesting times (April, August, and December), with the August harvest displaying the highest antioxidant activity [28]. The overall result of this study further revealed a positive and significant correlation between the total flavonoid content and the DPPH (r = 0.926), ABTS (r = 0.923), and FRAP (r = 0.734) antioxidant capacities (Table 1). Similar to this finding, Ouerghemmi et al. reported a higher correlation between the total flavonoids extracted from the flower of *Ruta chalepensis* and its antioxidant capacity [29].

### 2.3. Antimicrobial Activity of A. venetum and A. hendersonii Extracts

The antimicrobial activities of the total flavonoids from the two *Apocynum* species were significantly different (*p* < 0.05) and antagonistic against all three microbial strains compared to the control, as shown in Figure 2A. The zones of inhibition diameters for *A. venetum* extracts against *E. coli*, *MARS*, and *A. niger* were 11.20 mm, 9.81 mm, and 7.96 mm, while that of *A. hendersonii* were 17.80 mm, 13.99 mm, and 10.09 mm, respectively (Figure 2B). Compared to *A. venetum*, the *A. hendersonii* extract exhibited a larger inhibition zone, indicating better antimicrobial activities against all three strains, consistent with the results of Gao et al. [21]. The leaves of *Apocynum* L. collected from the Ili River Valley region of Xinjiang demonstrated a relatively higher total flavonoids content and bioactivity on the microbes (*A. niger*, fungi; *E. coli*, gram-negative bacteria, and *MARS,* gram-negative bacteria).

The extracts from both *Apocynum* species showed comparably higher activity against bacterial strains, having exhibited higher inhibition zones than the fungi (Figure 2C). The overall result thus revealed the medicinal and economic value of *Apocynum* plants as a promising antibacterial agent and could be a potential candidate in pharmaceutical industries. However, it remains to be seen whether there was a specific inhibition mechanism between the extracts of *A. venetum* and *A. hendersonii* against the three mentioned microbial strains. Since the membrane structure and composition of the three strains were significantly complex and different [30], the growth inhibition of strains is due to the abnormal changes in the morphology caused by the extracts, extravasation of intracellular material, or the inability of the cells to selectively control the intracellular transport of nutrients and metabolites, needs to be studied by further scanning electron microscopy.

### 2.4. Metabolomic Profiling of the Apocynum Species

The total ion current (TIC) and multi-peak detection plots for the *A. venetum* and *A. hendersonii* quality control (QC) samples are shown in Appendix A. The overlap of the TIC plots of the three QC samples indicated a complete overlap of the TIC plots of the metabolites, which demonstrated the instrumental stability and good reproducibility of the metabolite assays for the same samples at different time points. The metabolite multi-peak detection plot in MRM mode shows that the *A. venetum* and *A. hendersonii* contain multiple flavonoids, where each differently colored mass spectral peak represents one metabolite detected. The flavonoid metabolites in the leaves of *A. venetum* and *A. hendersonii* were studied based on UPLC-ESI-MS/MS and a metabolite database. A total of 83 flavonoid metabolites were identified, including 27 flavonols, 13 flavones, 13 flavanones, 10 catechin derivatives, 8 isoflavones, 5 anthocyanins, 4 proanthocyanidins, and 3 dihydrochalcones (Appendix A). The thermograms of all the flavonoid metabolites in *A. venetum* and *A. hendersonii* are shown in Figure 3A after homogenization, which indicated significant differences in the flavonoid metabolite contents. However, the compositions of the respective flavonoid metabolites were essentially the same. Cluster analysis further confirmed this, revealing a clear distinction between *A. venetum* and *A. hendersonii*. Moreover, it was found that more than half of the flavonoid metabolites in *A. venetum* were higher than in *A. hendersonii*.

### 2.5. Orthogonal Projection to Latent Structures–Discriminant Analysis (OPLS-DA) and Principal Component Analysis (PCA) for A. venetum and A. hendersonii

Orthogonal projection to latent structures-discriminant analysis (OPLS-DA) is an effective method for identifying differential metabolites as it maximizes the distinction between different groups [31]. Q^2^ is an essential parameter for evaluating the model in OPLS-DA, where a value of Q^2^ greater than 0.9 indicates that the model is a good one [31]. In this study, the OPLS-DA model was used to compare the flavonoid metabolite content of *A. venetum* (AV) and *A. hendersonii* (AH) samples (R^2^X = 0.945, R^2^Y = 998, Q^2^ = 0.984; Figure 3B,C). The Q^2^ values exceeded 0.9 in all control groups indicating the model’s stability and reliability [32] and can be used to further screen different flavonoid metabolites.

The principal component analysis (PCA) revealed the intrinsic structure between multiple variables, with PC1 and PC2 accounting for 49.7% and 15.4%, respectively. There was a clear separation between *A. venetum* and *A. hendersonii,* as shown in the PCA scoring plots (Figure 4A), with the replicates clustered closely together, demonstrating the reliability and reproducibility of the experiment.

### 2.6. Screening, Functional Annotation, and Enrichment Analysis of Differential Flavonoid Metabolites

To clarify the differences in flavonoid metabolites between *A. venetum* and *A. hendersonii* in greater depth, differential flavonoid metabolites were screened for each control group by combining the OPLS-DA model of ploidy variation and variable importance (VIP) values. Screening criteria included the fold change (≥2 or ≤0.5) and VIP ≥ 1 [31,32]. The results are shown in Table 2. In brief, there were 24 significantly different flavonoid metabolites (15 down-regulated and 9 up-regulated) between *A. venetum* and *A. hendersonii*.

The Kyoto Encyclopedia of Genes and Genomes (KEGG) database serves as the primary public database of metabolic pathways that can be used to study metabolite accumulation and gene expression information networks. In the present study, we enriched the differential metabolites of *A. venetum* and *A. hendersonii* and grouped them into different pathways, with the main pathways presented as bubble plots (Figure 4B, Appendix A). For the most notable, the metabolic pathways, including “flavonoid biosynthesis”, “ubiquinone and other terpenoid-quinone biosynthesis”, “biosynthesis of various secondary metabolites”, and “glycolysis/gluconeogenesis” were significantly upregulated in *A. venetum* compared to *A. hendersonii* (*p* value < 0.05). The correlation analysis of the first 37 different flavonoid metabolites between *A. venetum* and *A. hendersonii* is shown in Figure 4C, which visually demonstrates the intensity of potential chemical biomarkers between *A. venetum* and *A. hendersonii*.

### 2.7. Differential Flavonoid Metabolites between A. Venetum and A. Hendersonii 

The differential metabolites of *A. venetum* and *A. hendersonii* were grouped into eight different categories (Table 2). There were six major classes of flavonoid metabolites that were higher in *A. venetum* than in *A. hendersonii*, including isoflavones, flavonols, anthocyanins, proanthocyanidins, dihydrochalcones, and catechin derivatives. Only two major flavonoid metabolites, including flavones and flavanones, were lower in *A. venetum* than in *A. hendersonii* (Figure 4D). The overall result revealed that the activities exhibited by *A. hendersonii* were contributed to more by its rich flavone and flavanones, which were significantly higher than in *A. venetum*. Conversely, the activities exhibited by *A. venetum* were apparently due to its rich flavonols, which were comparably higher than the *A. hendersonii*. This constituent variation might be the reason for the species’ differential antimicrobial and antioxidant activities. Three notable flavones, vitexin, orientin, and isoorientin, as well as the flavanone eriodictyol, were found to be higher in *A. hendersonii*, which might be the reason for its better antimicrobial property. Vitexin, for example, has long been used as an antibacterial. It was recently demonstrated to play a critical role in modulating *S. aureus* surface hydrophobicity by aggregation forming biofilm and pathogenesis in a host organism. It also exhibited a protective response in *S. aureus*-infected macrophages via modulation of cytokine expression at protein and mRNA levels [33]. Orientin was reported to have displayed the best antibacterial effect against *Mycobacterium tuberculosis*, H37Rv [34]. Isoorientin and vitexin isolated from *Rumex cyprius* have shown promising activity against *Syncephalastrum racemosum* and *Streptococcus pneumoniae* at minimal concentrations of 0.98 and 1.95 μg/mL, respectively [35].

Conversely, the better antioxidant activity displayed by *A. venetum* might have been due to its relatively higher flavonols content. Flavonols such as quercetin and its derivatives are used in many pharmaceutical products as antioxidants [22]. Myricetin is shown to possess antitumor, anti-obesity, anti-inflammatory, and protection against cardiovascular and neurological ailments [36].

## 3. Materials and Methods

### 3.1. Plant Material

*A. venetum* and *A. hendersonii* were planted at the field of Ili Agricultural Institute, Xinjiang Uygur Autonomous Region, China (81°15′ E, 43°57′ N, altitude 656 m) in April of 2021. The leaves at the first 30 cm of the tender branches were selected, and samples were collected during four different periods (June, July, August, and September). The sampled leaves were then dried in an oven at 70 °C, ground into powder in a wall breaker, sieved through 160 mm meshes, and kept at −80 °C until analysis.

### 3.2. Extraction and Determination of Total Flavonoids

A modified approach [22] was adopted for the extraction. *Apocynum* L. leaf powder (5 g) was extracted with 100 mL methanol at 80 °C for 2 h. The supernatant was filtered with Whatman No.1 filter paper and dried with a rotary evaporator to obtain the dried extract powder, which was stored at 4 °C in the dark.

According to the method [22] with slight modification, the content of flavonoids was determined quantitatively by a spectrophotometer. In brief, 0.3 mL of 5% NaNO_2_ and 0.3 mL of 10% AlCl_3_ were added to 2 mL of the extract solutions and mixed, followed by 6 min incubation and the addition of 4 mL of 4% NaOH. The solution was then diluted to 10 mL with methanol and allowed to stand for 15 min before measuring the absorbance at 500 nm using a spectrophotometer (UV-3000, Mapada Instruments Inc., Shanghai, China). The total flavonoid content was calculated from a calibration curve of rutin as the standard (0–80 mg·mL^−1^) according to the above steps and expressed as rutin equivalents per gram of dry extract. All the procedures were repeated three times.

### 3.3. Antioxidant Assays 

The DPPH radical scavenging activity of the different extracts was performed following the method reported [37,38] with slight modifications. 2.5 mL of the extract was mixed with 1 mL of 0.3 mM DPPH in a 95% ethanol working solution and reacted for 30 min in the dark at room temperature, and the absorbance was measured at 518 nm. The DPPH radical scavenging ability was expressed as inhibition rate and calculated using the formula: (A_control_–A_sample_)/A_control_ × 100%, where A_control_ is the absorbance of the control solution (without sample), and A_sample_ is the absorbance of the sample solution.

The ABTS radical scavenging activity of the extracts was conducted according to the method described by Ghasemzadeh et al. [39] with slight modifications. A total of 950 μL of ABTS working solution was added to 50 μL of the extract, and the absorbance was measured at 405 nm after 6 min of reaction at room temperature in the dark. The free radical scavenging activity was calculated as a percentage of inhibition using similar formula as DPPH, where A_control_ is the absorbance of ABTS working solution without sample and A_sample_ is the absorbance of the sample solution. The experiments were all repeated three times, and the average values were taken.

The FRAP assay was conducted according to the method described by Yi et al. [40] with slight modifications. FRAP working solution consisted of 300 mM acetate buffer at pH 3.6, 10 mM TPTZ, and 20 mM FeCl_3_–6H_2_O in a 10:1:1 ratio, which was preheated to 37 °C before use. FRAP working solution (2.85 mL) was mixed thoroughly with 150 μL of *Apocynum* leaves extract and reacted for 10 min at room temperature under dark conditions, and the absorbance was measured at 593 nm. A calibration curve of Fe^2+^ was used to calculate the results. The reducing power of the methanolic extract of *Apocynum* leaves was expressed as µmol·ml^−1^. The higher the absorbance, the stronger the reducing power.

### 3.4. In Vitro Antimicrobial Assay

The extracts with the highest flavonoid contents (August harvest samples) for each species were selected for antibacterial assay and subsequent analysis. The in vitro inhibition assay was tested using the procedure described by Gao et al. [21]. The fungi, *Aspergillus niger* (ATCC 33591, Manassas, VA, USA) single colonies grown in Potato Dextrose Agar (PDA) solid medium (4.0% PDA, Solarbio, Beijing, China), were picked out and dispersed in 10 mL of fresh potato dextrose broth (2.5% PDB, Solarbio) and incubated at 28 °C, 200 rpm for 8 h. *Escherichia coli* (ATCC 25922) and Methicillin-resistant *Staphylococcus aureus* (ATCC 11632) were cultured in Luria Bertani (LB) solid medium (1% NaCl, 0.5% yeast extract, 1% tryptone and 1.5% agar, pH 6.8) or fresh LB liquid medium at 37 °C, and other conditions were the same as the fungi. The enriched microorganisms were then diluted to the same concentration (approximately 1.0 × 10^7^ CFU/mL) and then subjected to inhibition tests. A total of 100 μL of *A. niger* suspension was applied uniformly on PDA solid medium with sterilized cotton swabs, and 100 μL of *E. coli* and *MRSA* suspension was applied uniformly on LB agar plates.

A mixed solution (70% saline + 20% PEG-4000 + 10% DMSO) was added to the crude powder of the *Apocynum* extract to form a suspension of 20 mg/mL. All the liquid solutions were filtered through a 0.22 μm sterile filter membrane to remove bacteria, and all processes were performed under aseptic conditions. A total of 20 μL of the 20 mg/mL *Apocynum* L. leaf extract solution and control (co-solvent, 70% saline + 20% PEG-4000 + 10% DMSO) were separately used to soak filter paper (6 mm), and the paper was subsequently transferred onto the *A. niger*, *E. coli* and *MARS* media plates. The microorganisms were incubated at 30 °C (fungal) or 37 °C (bacterial) for 8 h. The transparent halo visible on the agar plate was considered the inhibition circle, and the diameter of the inhibition circle was measured and recorded with Vernier calipers. The average value of three biological replicates was taken for each strain.

### 3.5. Qualitative and Quantitative Analysis of Flavonoid Constituents of A. venetum and A. hendersonii

#### 3.5.1. Sample Preparation and Extraction

A total of 400 μL of chloroform and 0.6 mL of water: methanol (1:2 *v*/*v*) were added to 50 mg of *Apocynum* leaves extract [41], and ground for 2 min at 60 Hz using a grinder (JXFSTPRP-24/32, Shanghai Jingxin Industrial Development Co., Ltd., Shanghai, China) and centrifuged for 10 min at 13,000 rpm, 4 °C. The residues were re-extracted again using the same procedure. 200 μL of supernatant was evaporated and re-dissolved in 200 μL of water: methanol (18:7 *v*/*v*) solution containing 12 ng/mL of internal standard (L-2-chlorophenyl alanine), followed by centrifugation at 13,000 rpm at 4 °C for 5 min. A total of 100 μL of supernatant was filtered using a nylon syringe filter (BS-QT-013, 0.22 μm, Labgic Technology Co., Ltd., Beijing, China) and stored in sample bottles at −80 °C. All extraction reagents were pre-cooled at −20 °C before use, and three replicates were made for each sample. Quality control samples (QC) were prepared by mixing equal volumes of extracts from all samples, with each QC volume being the same as the sample.

#### 3.5.2. UPLC Conditions and ESI-Q TRAP-MS/MS

The *A. venetum* and *A. hendersonii* extracts were analyzed using a UPLC-ESI-MS/MS system (UPLC, AB ExionLC system; MS, Applied Biosystems 6500 Q TRAP, Framingham, MA, USA). The UPLC conditions were based on the reported method [42]. The analytical conditions were as follows: UPLC HSS T3 C18 (1.7 μm, 2.1 × 100 mm, Waters Technology Co., Ltd., MA, USA) chromatographic column was used. The mobile phases were phase A (0.1% formic acid-water solution) and phase B (acetonitrile). The gradient elution procedure was 95% solvent A and 5% solvent B within 10 min; the gradient was changed to 5% solvent A, 95% solvent B, and held for 1 min. Subsequently, the gradient was adjusted to 95% solvent A and 5% solvent B within 0.10 min and held for 2.9 min. The injection volume was 5 μL, the flow rate was 0.4 mL/min, and the column temperature was 40 °C. After UPLC, the effluent was alternately connected to an ESI-triple quadrupole-linear ion trap (QTRAP)-MS.

Mass spectrometry was also according to Wang et al. [42]. Linear ion trap (LIT) and triple quadrupole (QqQ) scans were performed on a triple quadrupole–linear ion trap mass spectrometer (Q TRAP). The API 6500 Q TRAP UPLC/MS/MS system is equipped with an ESI turbo ion-spray interface, operating in positive and negative ion modes, controlled by Analyst 1.6.3 software (AB Sciex, Framingham, MA, USA). The ESI source operation parameters were: an ion source, turbo spray, source temperature 600 °C; ion spray voltage (IS) 5500 V (positive ion mode)/−4500 V (negative ion mode); ion source gas I (GSI), gas II (GSII) and curtain gas (CUR) was set to 60, 50 and 35.0 psi, respectively; the collision gas (CAD) was high. Instrument tuning and mass calibration were performed in QqQ and LIT modes with 10 and 100 μmol/L polypropylene glycol solutions, respectively. QqQ scans were obtained in MRM experiments with collision gas (nitrogen) set to 5 psi. The dispersion potential (DP) and collision energy (CE) of the individual MRM jumps were obtained by further DP and CE optimization. A specific set of MRM transitions were monitored for each period based on the metabolites eluted.

The metabolites were analyzed using triple quadrupole mass spectrometry in multiple reaction detection (MRM) mode, in which the quadrupole first screens for precursor ions (parent ions) of the target substance while screening for ions corresponding to other molecular weight substances to initially exclude their interference; the precursor ions are broken into many fragment ions by collision chamber-induced ionization and filtered through the triple quadrupole to select individual fragment ions of the desired characteristics ions while excluding the interference of non-target ions, a step that gives better accuracy and reproducibility of quantitative results. After obtaining the mass spectrometry data from different samples, the peak areas were integrated. The peaks of the same substance in different samples were also corrected for integration. To compare the contents of each metabolite in different samples, the detected mass spectral peaks of each metabolite in different samples were corrected according to the information on retention time and peak shape of metabolites, which further ensured the accuracy of the qualitative and quantitative analysis.

#### 3.5.3. Multivariate Data Analysis 

The analyst 1.6.3 software (AB Sciex, Framingham, MA, USA) was used for processing the raw data signal. A log transformation of the raw metamorphic abundance was performed to normalize the data and achieve homogeneity of variance. Orthogonal projections to latent structures–discriminant analysis (OPLS-DA), cluster analysis, and principal component analysis (PCA) were performed using R (http://www.r-project.org/) according to the previously described approaches. The variable importance in projection (VIP) values of all metabolites in OPLS-DA was obtained using the first fraction. Metabolites that met both of the following criteria: (I) high level of confidence for paired comparisons (VIP ≥ 1); (II) minimum 2-fold change or maximum 0.5-fold change (fold change ≥ 2 or fold change ≤ 0.5) were selected as differential metabolites between *A. venetum* and *A. hendersonii*.

#### 3.5.4. Kyoto Encyclopedia of Genes and Genomes (KEGG) Annotations and Metabolic Pathway Analyses of Differential Metabolites

The differential metabolites were annotated and cataloged according to the KEGG database [43]. Enrichment results for metabolic pathways were defined by considering the pathways in the module where metabolites were involved in significant enrichment versus the background and determined by a hypergeometric test and *p*-value < 0.05 as a threshold, the smaller the *p*-value, the more significant the difference in that metabolic pathway.

### 3.6. Statistical Analysis

Statistical Package for the Social Sciences (SPSS v. 25.0; IBM Corp., Armonk, NY, USA) was used for data analysis of each treatment group. All the data were expressed as the mean ± standard deviation of three independent replicates. Differences were considered statistically significant when * *p* < 0.05, ** *p* < 0.01, *** *p* < 0.001 between treatment groups. Graphs were plotted using Origin 2018 (OriginLab Corporation, Northampton, MA, USA) software.

## 4. Conclusions

In this study, we investigated the optimal harvesting time of *Apocynum venetum* and *Apocynum hendersonii* for a maximal yield of total flavonoids, which are medicinally important constituents with higher bioactivity against clinically important diseases. A targeted metabolomics approach was also used to analyze the differences in the composition and relative contents of flavonoid metabolites of the two *Apocynum* L. species, and evaluate the antioxidant activities and the antibacterial activity against three microbial strains of *Escherichia coli*, *Staphylococcus aureus*, and *Aspergillus niger*. The results showed significant differences between the total flavonoid contents at the different harvesting periods, with August being the best. Bioactivity analysis revealed a striking difference between the two species extracts, with *A. venetum* displaying better antioxidant activities and *A. hendersonii* exhibiting better antibacterial effects. We thus investigated the likely reason for this remarkable difference. Our finding suggested that three notable flavones, vitexin, orientin, and isoorientin, which were significantly higher in *A. hendersonii,* might be responsible for the better antibacterial effect of the species extract. Flavonols such as quercetin, myricetin, and their derivatives, which have significant antioxidant activity, were conversely higher in *A. venetum*. The present work largely contributes to the understanding of the changes in the total flavonoid content of *Apocynum* L. leaves during different growth periods and proposes the best harvesting time according to the result and constituents’ bioactivities.

## Figures and Tables

**Figure 1 molecules-27-07343-f001:**
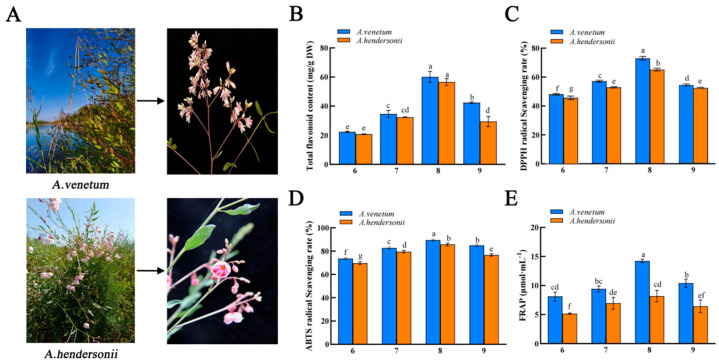
Morphology of *A. venetum* and *A. hendersonii*. (**A**) Total flavonoid content of *A. venetum* and *A. hendersonii* at different harvesting stages. (**B**) DPPH (**C**) and ABTS. (**D**) Radical scavenging activities, and FRAP. (**E**) The 6, 7, 8, and 9 represent the sampling periods June, July, August, and September, respectively. Histogram bars with different letters (a–f) are statistically different (*p* < 0.05).

**Figure 2 molecules-27-07343-f002:**
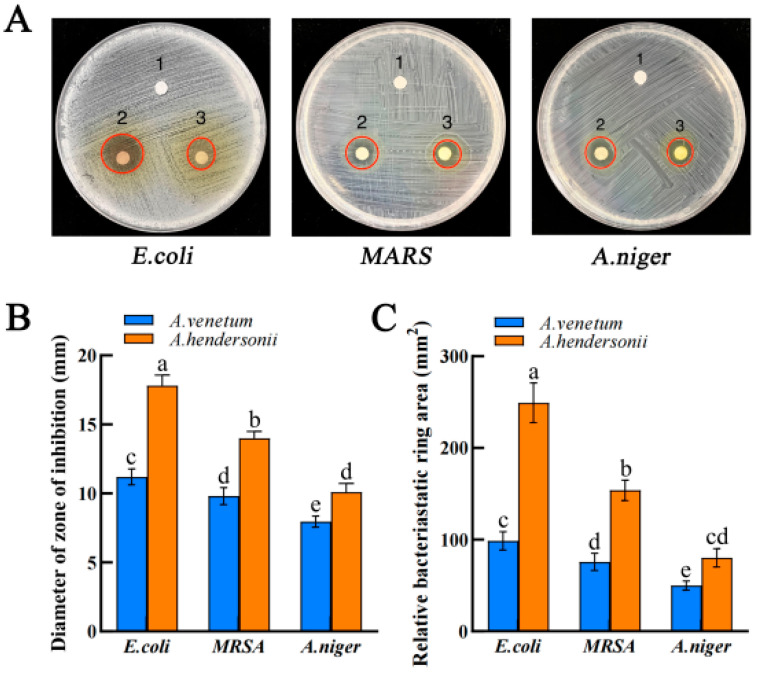
Digital images of inhibition zones of the *Apocynum* extracts against (**A**) Gram-negative *E. coli*, gram-positive *MRSA*, and fungal *A. niger* after treatment with different treatment groups: (1) Cosolvent; (2) *A. hendersonii* extract; (3) *A. venetum* extract, respectively. Diameter of the zone of inhibition. (**B**) The relative bacteriostatic ring area (**C**) was computed using $\pi r^{2}$. Data are presented as mean ± SD, *n* = 3. Histogram bars with different letters (a–e) are statistically different (*p* < 0.05).

**Figure 3 molecules-27-07343-f003:**
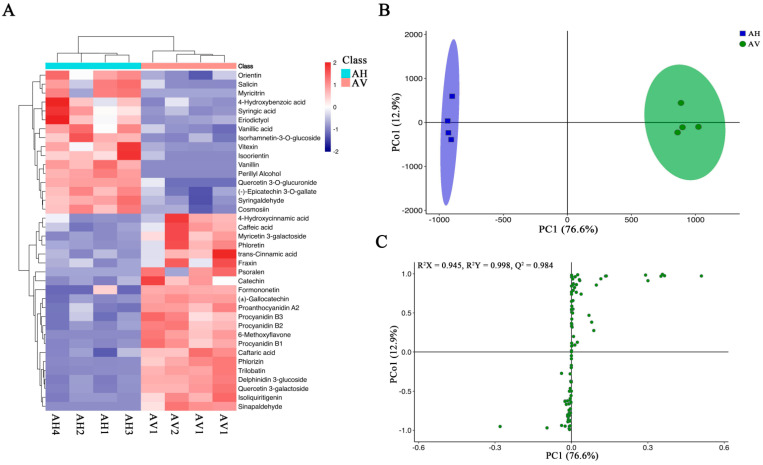
Heat map of the various flavonoid components and differential flavonoid metabolite analysis based on OPLS-DA. (**A**) a heat map of the various flavonoid components. (**B**,**C**) OPLS-DA differential metabolite analysis plots. AV means *A. venetum*, AH *A. hendersonii,* and the numbers 1, 2, 3, and 4 (as in AV1, AH1) stand for replicates numbers.

**Figure 4 molecules-27-07343-f004:**
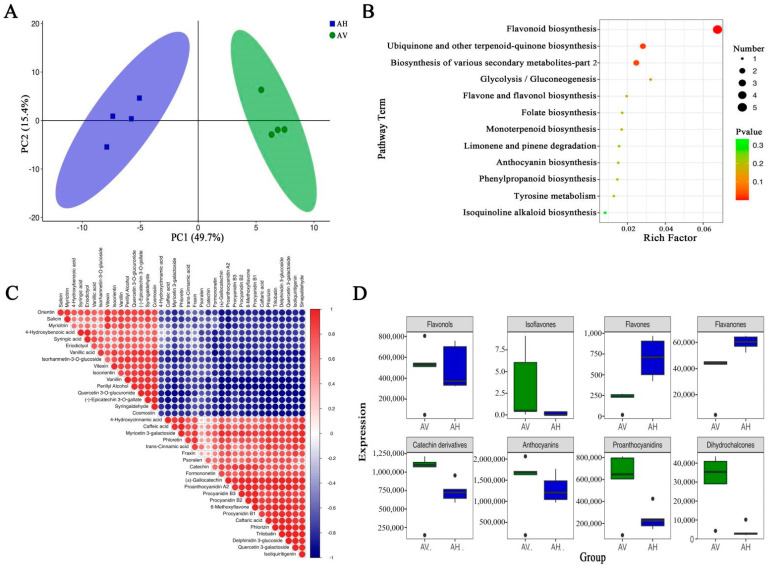
PCA score plot, KEGG enrichment analysis, correlation of flavonoid constituents and differentially enriched flavonoids of *A. venetum* and *A. hendersonii* (**A**–**D**). (**A**) PCA score plot. (**B**) KEGG enrichment of differential metabolites between AV and AH. (**C**) Correlations of AV and AH metabolites; the red and blue indicate positive and negative correlations, respectively. (**D**) Differentially enriched flavonoids of AV and AH. Each bubble represents a metabolic pathway. The colour of the bubbles represents the *p*-value for the enrichment analysis, with darker colours indicating higher levels of enrichment and larger bubbles indicating a more significant influence on the pathway.

**Table 1 molecules-27-07343-t001:** Pearson correlation coefficient of the *Apocynum* total flavonoid content and antioxidant capacities.

Antioxidant Assay	Correlation Coefficient
DPPH	0.926 **
ABTS	0.923 **
FRAP	0.734 *

**,* Significant correlation (*p* < 0.01, *p* < 0.05, respectively).

**Table 2 molecules-27-07343-t002:** A list of differential metabolites between *A. venetum* and *A. hendersonii*.

Compounds	Class	Average (AH)	Average (AV)	VIP	FC	log2 (FC)
Formononetin	Isoflavones	0.000	0.445	0.004	0.204	−2.292
6-Methoxyflavone	Flavones	0.000	0.629	0.006	0.000	−19.262
Vitexin	Flavones	320.056	13.003	0.122	24.614	4.621
Isoorientin	Flavones	59.500	3.760	0.052	15.823	3.984
Orientin	Flavones	138.480	44.195	0.068	3.133	1.648
Isoliquiritigenin	Flavanones	0.413	1.422	0.007	0.291	−1.783
Eriodictyol	Flavanones	191.398	52.812	0.079	3.624	1.858
Myricetin 3-galactoside	Flavonols	2479.295	27,784.466	1.156	0.089	−3.486
Cosmosiin	Flavonols	1430.489	262.041	0.247	5.459	2.449
Myricitrin	Flavonols	31.270	0.000	0.037	9.065	4.898
Quercetin 3-galactoside	Flavonols	48,353.573	296,554.439	3.655	0.163	−2.617
Quercetin 3-O-glucuronide	Flavonols	168,298.272	20,235.55	2.800	8.317	3.056
Isorhamnetin-3-O-glucoside	Flavonols	3351.903	479.293	0.387	6.993	2.806
Delphinidin 3-glucoside	Anthocyanins	32,306.137	188,173.056	2.911	0.172	−2.542
Proanthocyanidin A2	Proanthocyanidins	148.867	556.478	0.148	0.268	−1.902
Procyanidin B1	Proanthocyanidins	4123.930	38,822.697	1.370	0.106	−3.235
Procyanidin B3	Proanthocyanidins	431.080	2299.489	0.311	0.187	−2.415
Procyanidin B2	Proanthocyanidins	193,378.541	673,060.920	5.103	0.287	−1.799
Phloretin	Dihydrochalcones	12.653	115.965	0.071	0.109	−3.196
Trilobatin	Dihydrochalcones	112.871	1210.540	0.245	0.093	−3.423
Phlorizin	Dihydrochalcones	2479.889	35,922.788	1.340	0.069	−3.857
(-)-Epicatechin 3-O-gallate	Catechin derivatives	74.905	36.312	0.045	2.063	1.045
Catechin	Catechin derivatives	12,482.780	32,814.546	0.978	0.380	−1.394
(±)-Gallocatechin	Catechin derivatives	87,069.706	318,346.758	3.573	0.274	−1.870

Average (AV/AH): Average expression of *A. venetum*/*A. hendersonii* metabolites, VIP: Variable importance in the projection, FC: Fold change, Log^2^ (FC): Ratio of the mean expression of metabolites in the two sets of samples, positive values indicate upregulation, negative values indicate downregulation.

## Data Availability

Data is contained within the article and Appendix A, and also available from the corresponding author on request.

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
