# Peer review of "Total Flavonoids Extracts of Apocynum L. from the Ili River Valley Region at Different Harvesting Periods and Bioactivity Analysis"

_molecules, 2022, doi:10.3390/molecules27217343_

Round 1
Reviewer 1 Report (Previous Reviewer 1)
I suggest that the manuscript can be accept since the author has corrected all the errors identified by reviewers
Author Response
RESPOSES TO REVIEWER'S COMMENTS
Dear Professor,
Thank you very much for the time and in-depth evaluation of the proposed article. We appreciate all the comments. The whole text was checked for English usage issues and appropriately corrected. Grammatical and typographical errors were also checked and corrected. All changes made to the text are highlighted in red.
Thank you.
Sincerely,

Reviewer 2 Report (New Reviewer)
The research article entitled “Content Determination and Bioactivity Analysis of Native Flavonoids Extracted from Apocynum L. in the Ili River Valley Region” highlights the flavonoids extracted from the leaves of two Apocynum species collected from the Ili River Valley Region. The flavonoids from the leaf extracts were determined in consecutive months and the comparative analysis was done on the two species. The study has been designed and performed appropriately, however there are some observations which need to be addressed…
· Title: The title is confusing…..what is content determination?; what is native flavonoids?....the title should be rewritten..the title represents as if the flavonoids were taken and used in the study, but in reality the leaf extracts were used…
· Abstract: in view of the above observation the sentence “ the total content and bioactivity of native flavonoids were extracted from two Apocynum species leaves (Apocynum venetum and Apocynum hendersonii) collected from the Ili River Valley Region, and their bioactivities were investigated.” Should be corrected and rewritten, it is wrong to say bioactivity were extracted…
Purpose: my general question is the purpose of collection of leaves in consecutive months (June till September).. “ in my opinion it is wrong to say different Harvesting periods” it is a collection not a routine exercise…
Rewrite the sentence “The two species' leaves extracts demonstrated strong bioactivity, which positively correlated with the total flavonoid contents.” It should be leaf extracts from two species demonstrated…..”
Rewrite the sentence “These constituents' variability might be the reason for the different bioactivities displayed by the two species”. It should be “The Variability in these metabolites/constituents might be the reason…
· Introduction: this part is written appropriately.
· Material and methods: I was going through this section and realized that there is no mention in the paper about the comparative analysis between the four months. If the codes AV 1-4 and AH 1-4 represent the months then it should be mentioned in the material methods or if we look at the figure of correlation analysis it mentions in the legend repeats (these must be replicates). The months are mentioned in only figure 1 and represented as 6,7,8,9 meaning that the total flavonoid content assay and other antioxidant assays were the only ones performed for all months. Was the metabolomic analysis performed for all months? These things should be mentioned in materials and methods so that the readers don’t find it hard to follow..
· Results: The results part is nicely compiled and presented..
Correct the sentence “The extracts from both the two Apocynum species….” To “ The extracts from both Apocynum species…” or “ the extracts from two Apocynum species…”
Line 177: Use “was” instead of “were”
Line 178: Use “is” instead of “were”.
· Discussion: The discussion part is according to the observations made during the study and the cited references are appropriate.
Overall, the study does give novel information about the differential behaviour of two species and the influence of collection periods. However the paper needs some clarity in methodology section as mentioned above. There are some grammatical errors that should be corrected
Author Response
RESPOSES TO REVIEWER'S COMMENTS
Dear Professor,
Thank you very much for the time and in-depth evaluation of the proposed article. We appreciate all the comments. All the highlights were corrected, and all corrections are indicated in red in the text. We provided the pointwise response to the comments below:
Reviewers' comments and suggestions:
Title: The title is confusing…..what is content determination?; what is native flavonoids?....the title should be rewritten..the title represents as if the flavonoids were taken and used in the study, but in reality the leaf extracts were used…
Response:The title was modified according to the reviewer's recommendations.
Title: Total Flavonoids extracts of Apocynum L. from the Ili River Valley Region at Different Harvesting Periods and Bioactivity Analysis
Abstract: in view of the above observation the sentence “ the total content and bioactivity of native flavonoids were extracted from two Apocynum species leaves (Apocynum venetum and Apocynum hendersonii) collected from the Ili River Valley Region, and their bioactivities were investigated.” Should be corrected and rewritten, it is wrong to say bioactivity were extracted…
Purpose: my general question is the purpose of collection of leaves in consecutive months (June till September).. “ in my opinion it is wrong to say different Harvesting periods” it is a collection not a routine exercise…
Rewrite the sentence “The two species' leaves extracts demonstrated strong bioactivity, which positively correlated with the total flavonoid contents.” It should be leaf extracts from two species demonstrated…..”
Rewrite the sentence “These constituents' variability might be the reason for the different bioactivities displayed by the two species”. It should be “The Variability in these metabolites/constituents might be the reason…
Response: The suggestions made to the abstract were corrected, and the changes were highlighted.
In the current study, the total content from two Apocynum species leaves (Apocynum venetum and Apocynum hendersonii) collected from the Ili River Valley Region were extracted, and their bioactivities were investigated. The results showed a significant variation in the total flavonoid contents in the leaf samples collected at different periods (June, July, August, and September), with the highest content in August (60.11 ± 0.38 mg RE/g DW for A. venetum and 56.56 ± 0.24 mg RE/g DW for A. hendersonii), and the lowest in June (22.36 ± 0.05 mg RE/g DW for A. venetum and 20.79 ± 0.02 mg RE/g DW for A. hendersonii). The total flavonoid content was comparably higher in A. venetum than in A. hendersonii. Leaves extracts from the two species demonstrated strong bioactivity, which positively correlated with the total flavonoid contents. The anti-oxidative activity of A. venetum was higher than that of A. hendersonii in tandem with its higher flavonoid contents; the antibacterial activity however, was conversely opposite. Furthermore, a total of 83 flavonoid metabolites were identified in the two species based on UPLC-ESI-MS/MS, out of which 24 metabolites were differentially accumulated. The variability in these metabolites might be the reason for the different bioactivities displayed by the two species. The present study provides insight into the optimal harvest time for Apocynum species planted in the major distribution area of the Ili River Valley and the specific utilization of A. venetum and A. hendersonii.
Material and methods: I was going through this section and realized that there is no mention in the paper about the comparative analysis between the four months. If the codes AV 1-4 and AH 1-4 represent the months then it should be mentioned in the material methods or if we look at the figure of correlation analysis it mentions in the legend repeats (these must be replicates). The months are mentioned in only figure 1 and represented as 6,7,8,9 meaning that the total flavonoid content assay and other antioxidant assays were the only ones performed for all months. Was the metabolomic analysis performed for all months? These things should be mentioned in materials and methods so that the readers don’t find it hard to follow..
Response: The materials and method were modified accordingly. As observed, the comparative analysis between the four months was used for the total flavonoid determination and antioxidant assay. For subsequent analysis, we selected the best collection, which is the August sampling. The detail is now added to the text and indicated in red. AV 1-4 and AH 1-4 represent the replicates and are corrected in the text as suggested.
Results: The results part is nicely compiled and presented..
Correct the sentence “The extracts from both the two Apocynum species….” To “ The extracts from both Apocynum species…” or “ the extracts from two Apocynum species…”
Line 177: Use “was” instead of “were”
Line 178: Use “is” instead of “were”.
Response: The modifications suggested in the Results section were corrected. The whole text was checked for grammatical and typographical errors, as advised. All the changes made are highlighted in red in the text.
The extracts from both Apocynum species showed comparably higher activity against bacterial strains, having exhibited higher inhibition zones than the fungi (Figure 2C). The overall result thus revealed the medicinal and economic value of Apocynum plants as a promising antibacterial agent and could be a potential candidate in pharmaceutical industries. However, it remains to be seen whether there was a specific inhibition mechanism between the extracts of A. venetum and A. hendersonii against the three mentioned microbial strains. Since the membrane structure and composition of the three strains were significantly complex and different [30], the growth inhibition of strains is due to the abnormal changes in the morphology caused by the extracts, extravasation of intracellular material, or the inability of the cells to selectively control the intracellular transport of nutrients and metabolites needs to be studied by further scanning electron microscopy.
Thank you.
Sincerely,
This manuscript is a resubmission of an earlier submission. The following is a list of the peer review reports and author responses from that submission.
Round 1
Reviewer 1 Report
This present study investigated the relationship between native flavonoids extract of Apocynum species and harvest time, the bioactivity of native flavonoids extract was analyzed by antioxidant and antimicrobial experiments. The research optimizes the harvest time for Apocynum L. and provides a theoretical basis for its wider application. However, this manuscript has several issues that need to clarify in further, and the discussion of experimental results is not scientifically sound, it should be made a major modification before considering to accept.
1 It is well documented that Apocynum L. has strong hypotensive activity, why select to test antioxidant and antimicrobial activity, which is very easy and common, but less for research meaning and values, even is inconsistent with the results in vivo in most cases,
2 There are more redundant descriptions relative to experimental results in the abstract, it needs to be reorganized and simplified.
3 Line 101 of page 3, “…… and these were lower than what was obtained in this study”, how much native flavonoid extract was obtained at the same harvest time in this study? The specific data should be provided.
4 In Figure 1. B, C, D, E, F, What is the meaning of numbers 6, 7, 8, and 9 on the horizontal axis? It need mark it.
5 The annotations in Figure 2 are incomplete, there are no subtitles for B and C, it need provide.
6 “In Antioxidant Assays”, the data only compared the antioxidant activity of Apocynum L. at different harvesting stages, what is the positive control in the three experiments? The evidence of the conclusion in line 139 that “both A. venetum and A. hendersonii leaf extracts had potentially excellent antioxidant activity” is insufficient, more data should be provided to validate the results.
7 Line 163-165 of page 4, “Moreover, the extracts from both the two Apocynum specie…… differences in microbial membrane structure and composition.”, the data and references presented here are insufficient to support this conclusion, which should be supplemented with more data or references.
8 The description of the subtitle in Figure 3 is unclear, and the meaning of AH and AV should be annotated.
9 Section 2.6 only lists the data in Figure 4, there is a lack of analysis about them, which needs additional analysis and discussion in this section.
10 The manuscript only displays 4 biological replicates for metabolite analysis is not enough, more biological replications are needed to prove the reliability of these results.
Author Response
Dear Professor,
Thank you very much Sir for the time and in-depth check of the proposed article. We appreciated all the comments and attended all suggestions and corrections put forth. Please find attached the responses to the comments. All changes are highlighted in red in the text.

Reviewer 2 Report
Dear authors,
Reading your manuscript, it was enjoyable. It's intriguing to note that your results outperform any previously published data. Despite the fact that the antimicrobial capability is the opposite, the investigations revealed which extract appears to have a greater antioxidant capacity and which had greater antimicrobial activity. This underlines the significance of extract characterization in order to identify the chemicals that are responsible for various extract properties.
Please find few suggestions for changes to your text that you might want to make:
Line 94: What data is used for Figure 1C? What do you mean with as a whole in the Figure label? How it is different from Figure 1B?
Lines 170-172: Figure label is incomplete. There is no description for Figures 2B and 2C given.
Lines 275-278: Were the leaves taken from the same plant every month? Or were they taken from different plants? Did the plant vigor changed after removing the leaves?
Line 299: in the dark
Author Response

(The authors gave the same response as above.)

Reviewer 3 Report
Manuscript Title: Content Determination and Bioactivity Analysis of Native Flavonoids Extracted from Apocynum L. in the Ili River Valley Region
1- Abstract needs to be enhanced it has to address briefly and describe background, aim, object, procedure, results and important findings of novelty accordingly.
2- The content and structure of the presentation is confusing and needs to be rearranged and rewritten. Introduction lacks clarity, add few more literatures.
3- Modify the introduction section so that the readers could understand the flow of the research.
4- The conclusions should be pointed out with the major new findings described. In a present version of the paper, the conclusions are too general and just results rewritten.
5- Authors should be describing the morphology of the target species and highlights the main differences between them. The photo in figure 1 is not clear or enough to show the differences.
6- The main problem in this research is to use only two types of bacteria and one type of fungi to measure the antimicrobial activity, authors could be use at least 2 gram+ species and 2 gram- species and 2 fungi to be able to judge the antimicrobial activity of the plant extract.
Author Response

(The authors gave the same response as above.)

Round 2
Reviewer 1 Report
1. There are still some serious grammatical mistakes in the abstract, for example, “The anti-oxidative activity of A. venetum was higher than that of A. hendersonii in tandem with its higher flavonoid contents; the antibacterial activity was however, was conversely opposite.”, “was” should be deleted.
2. Line 52-63 of page 4, the additional content in the introduction is neither related to the effect of harvest time on the natural flavonoid content in Apocynum L., nor the antioxidants and anti-bacterial activity studied in this manuscript. The research advances related to these two aspects should be provided.
3. In “2.3. Antimicrobial activity of A. venetum and A. hendersonii extracts.”, only by the filter paper diffusion experiments cannot be proved that Flavonoid extracts in Apocynum L. have a stronger antibacterial activity against bacteria than fungi caused by “the differences in microbial membrane structure and composition.” Scanning electron microscopy experiments should be provided to support this conclusion
Author Response
RESPOSES TO REVIEWERS COMMENTS
Dear Professor,
Thank you very much Sir for the time and in-depth check of the proposed article. We appreciated all the comments and attended all suggestions and corrections put forth. All changes are highlighted in red in the text. We provided the pointwise response to the comments below:
Reviewers’s comments and suggestions:
1、There are still some serious grammatical mistakes in the abstract.
Response: The entire abstract was modified, and the grammatical errors were corrected. All the addition or changes to the text were highlighted in red.
Abstract: In the current study, the total content and bioactivity of native flavonoids were extracted from two Apocynum species leaves (Apocynum venetum and Apocynum hendersonii) collected from the Ili River Valley Region, and their bioactivities were investigated. The results showed a significant variation in the total flavonoid contents in the leaf samples collected at different harvesting periods (June, July, August, and September), with the highest content in August (60.11 ± 0.38 mg RE/g DW for A. venetum and 56.56 ± 0.24 mg RE/g DW for A. hendersonii), and the lowest in June (22.36 ± 0.05 mg RE/g DW for A. venetum and 20.79 ± 0.02 mg RE/g DW for A. hendersonii). The total flavonoid content was comparably higher in A. venetum than in the A. hendersonii. The two species' leaves extracts demonstrated strong bioactivity, which positively correlated with the total flavonoid contents. The anti-oxidative activity of A. venetum was higher than that of A. hendersonii in tandem with its higher flavonoid contents; the antibacterial activity however, was conversely opposite. Furthermore, a total of 83 flavonoid metabolites were identified in the two species based on UPLC-ESI-MS/MS, out of which 24 metabolites were differentially accumulated. These constituents' variability might be the reason for the different bioactivities displayed by the two species. The present study provides insight into the optimal harvest time for Apocynum species planted in the major distribution area of the Ili River Valley and the specific utilization of A. venetum and A. hendersonii.
2、Line 52-63 of page 4, the additional content in the introduction is neither related to the effect of harvest time on the natural flavonoid content in Apocynum L., nor the antioxidants and anti-bacterial activity studied in this manuscript. The research advances related to these two aspects should be provided.
Response: The whole introduction was checked and modified. The specific line mentioned and all others present were corrected. Further research advances were provided accordingly as requested.
Extracts from the species not only alleviated doxorubicin-induced cardiotoxicity through the AKT/Bcl-2 signaling pathway, but also showed significant hepatoprotective effects against carbon tetrachloride-induced hepatotoxicity [13, 14]. The characteristic made the plant adapt to not only drought and saline lands but also a suitable alternative to synthetic human drugs for treating heart disease, hepatitis, and hypertension [15, 16].
3、In “2.3. Antimicrobial activity of A. venetum and A. hendersonii extracts.”, only by the filter paper diffusion experiments cannot be proved that Flavonoid extracts in Apocynum L. have a stronger antibacterial activity against bacteria than fungi caused by “the differences in microbial membrane structure and composition.” Scanning electron microscopy experiments should be provided to support this conclusion.
Response: The specific part mentioned, and other sections of the Result and Discussion were checked and appropriately corrected. This article attempted to investigate the best harvesting time of the species for maximal flavonoid yield and, consequently, medicinal utilization. That is why the use of the antioxidant and antimicrobial activity tests to further supports the result. The medicinal and economic value of Apocynum plants as a promising antibacterial agent and could be a potential candidate in pharmaceutical industries. While, it remains to be seen whether there were specific inhibition mechanism between the extracts of A. venetum and A. hendersonii against the three mentioned microbial strains. Since the membrane structure and composition of the three strains were significantly complex and different, the growth inhibition of strains were due to the abnormal changes in the morphology caused by the extracts, extravasation of intracellular material, or the inability of the cells to selectively control the intracellular transport of nutrients and metabolites needs to be studied by further scanning electron microscopy.
All changes made to the result and discussion, were indicated using track.
2.3. Antimicrobial activity of A. venetum and A. hendersonii extracts
The antimicrobial activities of the total flavonoids from the two Apocynum species were significantly different (p < 0.05) and antagonistic against all three microbial strains compared to the control, as shown in Figure 2A. The zones of inhibition diameters for A. venetum extracts against E. coli, MARS, and A. niger were 11.20 mm, 9.81 mm, and 7.96 mm, while that of A. hendersonii were 17.80 mm, 13.99 mm, and 10.09 mm, respectively (Figure 2B). Compared to A. venetum, the A. hendersonii extract exhibited a larger inhibition zone, indicating better antimicrobial activities against all three strains, consistent with the results of Gao et al. [21]. Comprehensively, the leaves of Apocynum L. collected from Ili River Valley region of Xinjiang. demonstrated a relatively higher total flavonoids content and bioactivity on the typical microbials (A. niger, fungi; E. coli, gram negative bacteria and MARS, gram negative bacteria).
The extracts from both the two Apocynum species showed comparably higher activity against bacterial strains, having exhibited higher inhibition zones than the fungi, A. niger (Figure 2C). The overall result thus revealed the medicinal and economic value of Apocynum plants as a promising antibacterial agent and could be a potential candidate in pharmaceutical industries. While, it remains to be seen whether there were specific inhibition mechanism between the extracts of A. venetum and A. hendersonii against the three mentioned microbial strains. Since the membrane structure and composition of the three strains were significantly complex and different [30], the growth inhibition of strains were due to the abnormal changes in the morphology caused by the extracts, extravasation of intracellular material, or the inability of the cells to selectively control the intracellular transport of nutrients and metabolites needs to be studied by further scanning electron microscopy.
We appreciate every comments and suggestions by the reviewers. We also wish to express our gratitude to the Editorial team for the chance to respond to the comments.
We hope our manuscript will be accepted for publication by the Journal.
Thank you.
Sincerely,
